# HEATSNAP: A Hot Page-Aware Continuous Snapshots System for Virtual Machines in Web Infrastructure

Submission Id: 255

## Abstract

Snapshot technology is crucial for data protection and system recovery in virtualized environments, particularly with the growing need for continuous snapshots to maintain the integrity of long-running web-based and distributed applications. However, traditional snapshot methods often suffer from performance bottlenecks, and inefficient storage usage. These challenges are closely tied to the way memory pages are accessed during VM execution, where memory access patterns show significant disparities between frequently accessed "hot" pages and less-used "cold" pages.In this paper, we introduce HEATSNAP, a continuous snapshot system designed to address these issues by leveraging the uneven access frequencies of memory pages. HEATSNAP distinguishes between intensive hot pages and dirty pages, applying specialized snapshotting and storage strategies to optimize the handling of both hot and cold memory regions. This approach aims to optimize snapshot efficiency, minimize performance impact on the VM, and decrease storage costs.Our implementation of HEATSNAP on QEMU/KVM demonstrates significant improvements in VM performance loss, snapshot duration, and storage efficiency compared to existing methods, as evidenced by evaluations on common web and cloud-based workloads.

## 1 Introduction

In recent years, with the rapid evolution of cloud computing platforms, virtualization has become a core technology, widely applied in web-scale infrastructure. Through virtualization, these platforms achieve flexible resource allocation and management, enhance hardware utilization, reduce costs, and offer users elastic computing environments [12, 18]. To ensure compliance with Service Level Agreements (SLAs), cloud computing platforms must provide reliable data protection and recovery mechanisms. Snapshot technology has emerged as a crucial solution for this purpose, capturing the state and data of virtual machines and saving them as images for future recovery needs. By leveraging snapshot technology, cloud platforms can perform data backup, recovery, and migration operations with increased reliability and flexibility [8, 21, 29].

Continuous snapshot technology builds on traditional methods, enabling frequent snapshots to enhance data protection and recovery in large-scale, web-based environments. By employing continuous snapshots, cloud platforms can perform regular data backups, minimizing the risk of data loss and improving system resilience [8, 22, 27, 30]. This approach typically uses incremental backups, storing only modified data blocks, which conserves storage space and reduces backup time [11]. Additionally, continuous snapshots offer fast recovery options, enabling web platforms to restore services from specific time points with minimal disruption. Existing methods often use stop-copy [7, 31], pre-copy [30], and post-copy [13] techniques, combined with incremental mechanisms to store changes between snapshots.

| | Downtime | Duration | Perf. Loss | Size |
|---|---|---|---|---|
| Stop-copy | Long | very Short | None | Med |
| Pre-copy | Med | Long | Small | Big |
| Post-copy | very Short | Med | Med | Med |
| HEATSNAP | Short | Short | Small | Small |

**Table 1: Snapshot solutions comparison with HEATSNAP**

For virtual machine snapshots, key evaluation metrics include VM downtime, duration time, performance loss, and snapshot size. VM downtime signifies the period during which the virtual machine is paused for snapshotting, while duration time reflects the overall time required for the snapshot process. Performance loss quantifies the performance reduction experienced by the virtual machine (excluding downtime) during its operation due to snapshot activities, impacting runtime smoothness and user experience. Snapshot size indicates the volume of the stored snapshot file. Existing solutions exhibit certain limitations, as depicted in Table 1.

Stop-copy pauses the VM entirely during snapshot creation, leading to longer downtime but without extra overhead. Pre-copy iterates several times to transfer modified memory pages, extending the snapshot duration as new changes occur in each iteration. Post-copy saves the VM's basic state during downtime and transfers memory pages in the background, competing with other VMs for CPU and I/O resources, which can degrade performance. Furthermore, existing continuous snapshot methods often neglect storage costs, a critical factor for ensuring business continuity and availability.

This study presents HEATSNAP, a system that optimizes continuous snapshots by capitalizing on hot page characteristics. The approach involves addressing two key questions: (1) Identifying hot page features and associated technical challenges, and (2) Incorporating hot page traits in snapshot mechanism design.

Initially, we outline three features and challenges of hot pages during virtual machine execution to tackle the first question. Firstly, efficiently identifying hot pages from a large pool of memory pages is a challenge due to the concentrated distribution of hot and cold pages. Secondly, tracking the dynamic changes in hot pages and promptly responding to alterations pose significant challenges. Lastly, managing the fluctuating load of hot pages to maintain an optimal saving strategy presents an ongoing challenge.

Subsequently, we introduce HEATSNAP, a continuous snapshot system tailored to address the second question. Leveraging the Post-copy method, HEATSNAP implements a mechanism to preserve dirty pages separately as hot and cold pages. Specifically, during downtime, a select few hot pages are pre-saved to minimize page faults during virtual machine operation. Additionally, a dynamically incremental strategy adjusts storage tactics based on the virtual machine's load pressure to maintain a balance between real-time snapshot performance and virtual machine efficiency. Moreover, a

snapshot compression mechanism is devised to reduce snapshot storage expenditures.

To verify these concepts, we implemented a prototype on QEMU [25] / KVM [19] and conducted a series of experiments. In the process, we compared HEATSNAP's performance with simplified adaptations of existing techniques. Specifically, sRemus is a streamlined version of Remus [7] that employs the stop-copy method; sQE is a streamlined version of Quick-Eviction [10], utilizing pre-copy; and siConSnap is a streamlined version of iConSnap [13], following the post-copy strategy. This comparison demonstrated clear improvements in performance. HEATSNAP reduces snapshot duration by 17.4% and diminishes VM performance loss by 48.4% compared to siConSnap. Furthermore, storage costs and snapshot boot time are slashed by 46.7% and 72.2%, respectively, in contrast to original methodologies.

The contributions of this article are as follows:
- We summarize the characteristics of hot pages and analyze the technical challenges they pose to snapshot technologies.
- We present a case of snapshot system integrated with hot page management, offering an efficient solution for designing hot page-aware snapshot systems.
- We implemented a prototype called HEATSNAP, equipped with a set of effective techniques, on QEMU/KVM.
- Extensive experiments validate the effectiveness of HEATSNAP.

## 2 Background and Motivation

### 2.1 Single Snapshot

Most existing snapshot methods rely on stop-copy, pre-copy [7, 30, 31], or post-copy techniques [13–15]. In stop-copy, the VM is paused to capture its full state, followed by a complete state save, after which the VM is resumed. This method is commonly used in virtualization platforms like KVM [19], Xen [2], and VMware [16], but its extended downtime limits its usefulness in cloud environments.

Pre-copy was introduced to address this issue, running snapshot processes concurrently with VM operations to reduce downtime [4, 17]. Dirty pages are transferred in multiple iterations, with the VM pausing briefly at the end to flush remaining pages. This reduces downtime, but in write-heavy workloads, repeated dirty page transfers can prolong the total snapshot process.

Post-copy, on the other hand, delays memory page transfer [14]. After a brief pause to save device and CPU states, the VM resumes while pages are transferred in the background. Any modifications to unsaved pages trigger page faults, which forces the system to save the page before resuming the VM. Although this method minimizes downtime, background transfers compete with the VM for resources, which can degrade performance, especially in multi-VM environments with high write activity, impacting user experience during the snapshot process.

Current research aims to further process snapshot data to minimize the IO process overhead during snapshot periods. These research techniques include memory compression [5], duplicate page merging [24], and idle page detection [1, 6, 20]. The objective is to reduce persisting data time in the snapshot storage process by compressing data size and consolidating duplicate data. However, these approaches introduce substantial performance overhead

and impact the system's overall execution. Adaptive methods such as snapshot buffers and asynchronous writing [13] strive to balance CPU and IO resources to prevent excessive CPU utilization by the aforementioned methods. Nonetheless, their effectiveness diminishes under heavy workloads, potentially creating additional unnecessary overhead.

### 2.2 Continuous Snapshots

Currently, continuous snapshots predominantly employ an incremental snapshot mechanism to store the updated data since the last snapshot, minimizing duplicated data between consecutive snapshots [7, 9, 11, 31, 34, 37]. Recent research mainly concentrates on diminishing redundant data, deferring data availability, and delaying storage to reduce the size of the incremental data. This strategy enhances the efficiency of continuous snapshots and mitigates storage overhead.

Remus [7] specifically introduced an approach using pre-copy for the initial full snapshot and optimizing stop-copy with various techniques for subsequent incremental snapshots to achieve frequent snapshots. However, although this method facilitates high-frequency snapshots, it still involves notable downtime due to the necessity of copying altered memory pages during downtime. Egger et al. [9] aimed to reduce the snapshot size and duration by excluding memory pages recoverable from the hard disk during the stop-copy phase. Despite these efforts, the method results in relatively prolonged downtime, particularly for memory-intensive operations. iConSnap [13] implemented a lazy saving mechanism that tracks dirty page information using a bitmap between snapshots, delaying saving until the snapshot phase. While this technique lessens the snapshot size and virtual machine downtime, it impacts virtual machine performance and snapshot real-time capabilities.

To curb the storage expenses associated with continuous snapshots, Ta-Shma et al. [30] proposed discarding all VM states beyond the accessible window. This simplistic recycling strategy, however, leads to the loss of valuable information, rendering it impossible for the VM to revert to a previous state.

### 2.3 Hot page Characteristic and Challenge

Most mainstream virtual machine snapshot systems treat all memory pages uniformly, without leveraging the differing access frequencies of pages during VM operations. To address this, we analyzed hot page data across six workloads—Idle, Redis, 7zip, FIO, Memcached, and MPlayer—summarizing three key characteristics of hot pages, as shown in Figure 1.

**Concentrated cold and hot distribution.** The analysis shows that most memory pages in a snapshot are cold, with a small proportion being hot, and an even smaller fraction having medium access frequencies. Continuous snapshots primarily involve hot pages, requiring a more refined method to identify them. Efficiently locating hot pages at low cost from large memory sets is a key challenge.

**Rapid change.** Hot pages shift frequently in real-world workloads, driven by transient tasks. As virtual machine operations rapidly transition between tasks, hot page distributions change abruptly. Simple sampling methods based on page modification

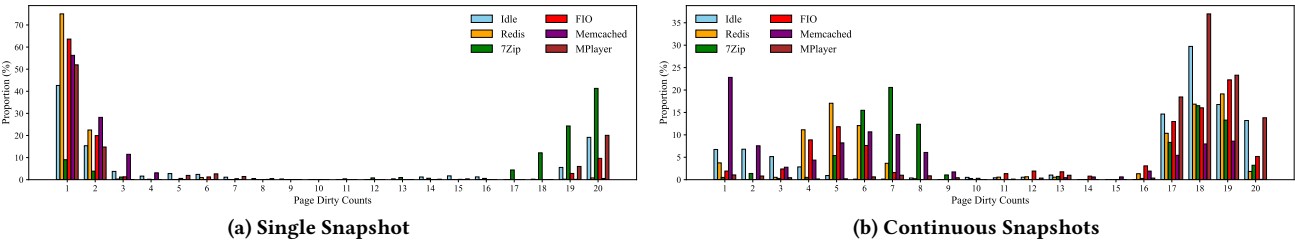



(a) Single Snapshot       (b) Continuous Snapshots



**Figure 1: Statistics of Hot pages in the Snapshots of VM Running Common Workloads**

frequency fail to capture these rapid shifts, making efficient hot page detection a major challenge.

**Large load fluctuations.** Hot pages display strong temporal locality, with high access rates during memory-intensive tasks and significantly lower rates during less demanding tasks. Adapting snapshot strategies dynamically to accommodate these fluctuations is critical for balancing snapshot efficiency and VM performance.

Table 1 outlines the limitations of existing snapshot methods, which fail to account for varying memory access patterns. This has motivated the development of a new system that leverages hot page characteristics to enhance snapshot performance, reduce downtime, and optimize storage with minimal overhead.

## 3 HEATSNAP Design

### 3.1 Overview

HEATSNAP is a swift, low-overhead, hot page-aware continuous snapshot system, as depicted in Figure 2. It comprises four key components: hot page detection, single snapshot, continuous snapshots, and snapshot compression mechanism. The hot page detection mechanism swiftly identifies hot pages and handles hot pages and cold pages differently during the snapshot process, thereby reducing snapshot duration. The continuous snapshot employs a dynamically incremental strategy to adjust the number of pages written to disk based on dirty page generation, ensuring a balance between performance and availability. The snapshot compression mechanism introduces two lossless techniques to eliminate redundant data in snapshots, thus reducing storage costs.

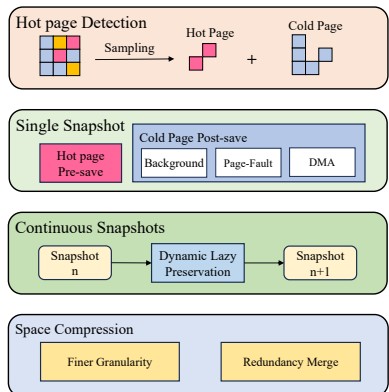

**Figure 2: HEATSNAP Overview**

### 3.2 Hot page Detection

We assert that an effective hot page detection method should exhibit four key characteristics: low overhead, locality, high accuracy, and high responsiveness. Low overhead necessitates that the method minimizes the burden on CPU and memory, particularly by reducing unnecessary sorting operations. Locality mandates that the method prioritize recent memory page accesses, assigning more weight to recent accesses to enhance hot page detection accuracy. Additionally, high responsiveness demands that the method swiftly detects alterations in hot pages.

We designed a dynamic working set based on recent page modifications, adjusting the set size and page weights to dynamically recalibrate the *hot page threshold*. The core steps of the algorithm are as follows:

*3.2.1 Definition of the Working Set and Page Weight Assignment.* The working set initially consists of the most recent modifications of a page. We assign weights to the pages in the working set, where the weights increase as the modification is more recent. The weights are distributed as follows:

$$\text{weight} = \left\{1, 2, 4, \ldots, 2^n\right\}$$

When a page is modified within the working set, its corresponding weight is incremented. Based on the accumulated modification history, we assign a *dirty page weight* to each page, ranging from 0 to $2^{n+1} - 1$.

*3.2.2 Initial Threshold for Hot page Identification.* The initial threshold is set at 9, meaning that pages with a weight of 9 or higher are classified as *hot pages*. This threshold is determined by the fixed working set size and the modification history of the pages.

*3.2.3 Increasing the Threshold and Heat Level.* When the system detects an excessive proportion of hot pages due to memory-intensive tasks, it increases the criteria for classifying a page as a hot page by expanding the working set.

For example, expanding the working set to 5 results in weights of 1, 2, 4, 8, 8, and the threshold is raised to 19. Under this configuration, a page must have been modified in at least three of the most recent five samples to qualify as a hot page.

As the heat level $n$ increases, the working set expands to $n + 3$, and the page weights follow:

$$\text{weight} = \left\{1, 2, 4, \ldots, 2^n, 2^{n+1}, 2^{n+1}\right\}$$

The threshold for hot page identification is then calculated as:

$$T = 3 \times 2^{n+1} - 1 - 2^{\left\lceil \frac{n+1}{2} \right\rceil}$$

Consequently, within the most recent $n + 3$ samples, a page must have been modified in all but the $(n+3)/2+1$-th most recent sample, or modified in all of the most recent samples, to be classified as a hot page.

*3.2.4 Heat Level Adjustment Mechanism.* The system dynamically adjusts the heat level through an "increase heat" or "decrease heat" mechanism, aiming to maintain the desired ratio of hot pages. If the system detects an oscillation in the heat level, such as "increase-decrease-increase" patterns, it enters a *slow adjustment phase*, in which the threshold is incremented or decremented by 1 unit at a time. This slow phase lasts for up to 10 samples before switching back to the fast adjustment phase for larger, more responsive changes.

This approach ensures that the system dynamically adapts to varying memory loads, efficiently managing the ratio of hot pages and minimizing system downtime.

## 3.3 Single Snapshot

A virtual machine snapshot is crucial for capturing the complete state of the virtual machine at a specific moment, encompassing the CPU, memory, disk, and other device states. HEATSNAP primarily emphasizes optimizing the preservation of memory states through efficient methods tailored to the memory page's hot or cold status.

Upon receiving a snapshot creation command, HEATSNAP initiates by pausing the virtual machine to lock its state precisely when the snapshot command is issued. Subsequently, it records the CPU state, disk state, and other device states. Saving the CPU state primarily involves preserving register values, while employing the Redirect-on-Write method for disk and device states, both of which are swift processes.

Following this, the memory state is archived. During this downtime, the primary task involves analyzing hot page data, storing hot pages in a buffer, and enforcing write protection on cold pages. Given that hot pages are a minority in the memory pool, this procedure is likewise rapid. Once completed, the virtual machine resumes operation, while a background thread continues saving the remaining pages. Throughout this phase, modifications to write-protected pages prompt page faults, inducing the passive preservation of altered pages and subsequent removal of their write protection. To ensure coherence between these actions, a dirty page bitmap is employed.

## 3.4 Continuous Snapshots

Continuous snapshots of a virtual machine involve a complete snapshot at the outset and subsequent modifications. Traditional incremental snapshot storage methods capture real-time dirty page data through Copy-on-Write (COW). However, this approach becomes problematic during peak virtual machine activity, as the heightened incremental storage exerts a substantial toll on the virtual machine's already burdened performance. In contrast, iConSnap introduces a lazily incremental method that monitors page modifications between two snapshots, deferring actual data storage until the following snapshot. While this method mitigates the performance impact of real-time saving, it compromises real-time and availability of snapshots. In contrast, HEATSNAP proposes a dynamically incremental strategy that achieves an optimal equilibrium

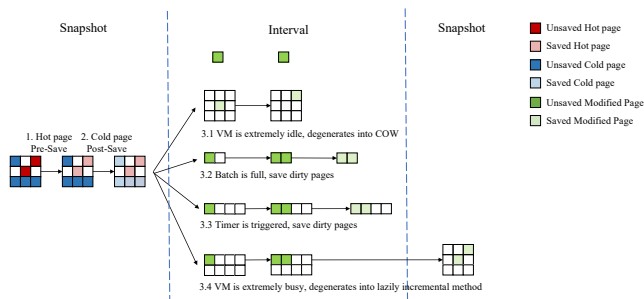

**Figure 3: Dynamically Incremental Strategy in Continuous Snapshots**

between immediate performance and overall system performance by dynamically adjusting to the current virtual machine workload.

As presented in Figure 3, HEATSNAP aggregates multiple modification records into batches and stores dirty page data in batch units. Adjustments are made to the batch size dynamically based on the virtual machine's workload: increasing the batch size during high load periods to mitigate incremental saving performance degradation and decreasing it during low load intervals to enhance real-time snapshot performance. A periodic timer supervises the batch status, triggering adjustments accordingly. If the batch fills before the timer expires, signaling high load, the batch size doubles to reduce disk write frequency and alleviate performance loss. Conversely, if the timer activates before the batch reaches capacity, indicating low load, the batch size halves to increase the write frequency and elevate snapshot real-time.

To prevent excessive fluctuation in batch size under extreme conditions, HEATSNAP sets limits on batch adjustments. In times of extensive idleness, the batch size progressively halves until reaching one, converging the continuous snapshot strategy to a conventional COW method with immediate saving for each modification. Conversely, during intensive virtual machine activity, the batch size continually doubles until reaching the upper threshold. In this scenario, the continuous snapshot method transitions into a lazily incremental method.

## 3.5 Snapshot Compression

Continuous snapshots result in a proliferation of snapshot files during long-term operations, leading to substantial increases in storage costs. To address this, conventional methods typically establish a time window, restricting the virtual machine state restoration period and eliminating data older than this window [26]. While existing optimizations aim to identify and preserve critical historical data, they do not fully resolve the issue of losing old snapshots. In response, we present two lossless snapshot compression techniques, fine-grained compression and redundancy merging, designed to reduce snapshot storage overhead without compromising data integrity.

**Fine-grained Extraction.** Throughout our discussions on memory hotspots, we have predominantly addressed the page as the smallest unit. However, within a hot page, modified data may only occupy a fraction of the entire page. Storing the entire hot page each time results in redundant data consumption and significant space wastage. Hence, we advocate for a finer granularity in storing

incremental data in continuous snapshots. By saving solely the actual modified data within the page, a notable reduction in snapshot space consumption is achieved, facilitating snapshot compression. The optimal granularity size selection is elaborated on in §5.2.4.

**Redundancy Merging.** Conversely, when dealing with multiple single snapshots and virtual machines, it is probable that identical virtual machines operate the same OS, libraries, and applications across various snapshots, while simultaneous virtual machines may run identical software. Consequently, numerous memory pages are identical. Exploiting this scenario, we exclusively save the actual data of duplicate pages once and assign references, in the format $vm\_id, snapshot\_id, pfn$, to replace redundant sections. Compared to storing complete 4kB page data, this triplet substantially minimizes data space, effectively eliminating redundancy. Utilizing hashing to identify duplicate data, we compute hash values of memory page data from recent snapshots for each virtual machine. When saving a new snapshot, if a matching hash result is present in the hash table, solely a reference to the corresponding memory page is retained; otherwise, the complete data is stored, and the hash result is incorporated into the hash table.

## 4 HeatSnap Implementation

We implemented HeatSnap using QEMU 2.5.0 and Linux 4.15.0, totaling 2980 lines of C code.

For the background save mechanism, a dirty page bitmap (`dirty_bitmap`) was devised to represent all pages requiring preservation. During the intervals between continuous snapshots, an `inc_bitmap` logs modified pages. During system downtime, these modified pages are transferred to the `dirty_bitmap` for protection, and the `inc_bitmap` is then reset. Meanwhile, a background process saves the dirty page data based on the updated `dirty_bitmap`, enabling the virtual machine to resume normal operation while the save process continues.

In terms of snapshot compression, we utilized a hash table based on open addressing to store hash values and related metadata, employing the MurmurHash3 algorithm with an initial size of approximately 1 million entries. Additionally, a B+ tree structure organized and indexed vast metadata, incorporating virtual machine ID, snapshot ID, and page frame numbers. Data in snapshot files were compartmentalized into the hash table, metadata index, and actual page data to enhance compression.

## 5 Evaluation

### 5.1 Experimental Setup

For our experimental framework, we utilized QEMU version 2.5.0. The virtual machine was configured with 2GB of memory and 2 vCPUs. Alpine 3.16, running kernel version 5.15.158-0-lts, served as the operating system within the virtual environment. Additionally, the host or physical machine ran on an Ubuntu 16.04 operating system, with a kernel version of 4.15.0. We further constrained the CPUs accessible to QEMU to 2 as to emulate an environment with restricted CPU resources.

**Workloads.** We selected the following workloads to evaluate our system's performance under various tasks relevant to web infrastructure:

(1) **Idle**: Measures system behavior when the virtual machine is inactive, providing a baseline for resource usage.
(2) **7zip**: Tests memory usage during data compression, a key operation for reducing bandwidth in web applications that handle large data transfers.
(3) **Redis**: Benchmarks an in-memory key-value store, widely used in web applications for caching and fast data access, using YCSB [3] as a workload generator.
(4) **FIO**: Simulates I/O-intensive tasks to assess storage performance, essential for web services that handle high volumes of data transactions.
(5) **Memcached**: Evaluates memory performance for caching, critical for reducing database load and latency in dynamic web applications, using mcperf [33] as a client.
(6) **MPlayer**: Streams a 1.5GB video to test the system's ability to handle continuous data reading, relevant for web platforms offering media streaming services.

**Metrics.** As mentioned earlier, we evaluate snapshot performance based on the following metrics: stop time, duration time, and performance loss. We compared HeatSnap with the following three systems:

(1) **sRemus**: A system that implements continuous snapshots using stop-copy and incremental methods (a streamlined version of Remus [7]).
(2) **sQE**: A system that implements continuous snapshots using pre-copy and incremental methods (a streamlined version of Quick-Eviction [10]).
(3) **siConSnap**: A system that implements continuous snapshots using post-copy and incremental methods (a simplified variant of iConSnap [13]).

In the following sections, we first test the proposed snapshot optimization methods to demonstrate their correctness and potential for efficiency improvements. Then, we compare the performance of various snapshot systems under different workloads using our proposed metrics.

### 5.2 Key Technique Analysis

*5.2.1 Evaluating Hot page Detection.* In this section, we evaluate the accuracy and overhead of the HeatSnap hot page detection algorithm. We define the accuracy as $\sum_{\text{detected}} / \sum_{\text{real}}$, where $\sum_{\text{detected}}$ is the number of hot pages identified by our algorithm, and $\sum_{\text{real}}$ represents the actual number of hot pages. The actual hot pages are determined by sampling every 30 seconds, counting accesses over 60 samples, and sorting these counts in descending order. We also compared HeatSnap with a traditional sorting-based method, denoted as HeatSnap+Rank. In this approach, we sample the number of accesses in the last 10 samples and select a specified proportion of hot pages based on this sorted list.

As depicted in Figure 4, our technique attains a hotspot detection accuracy remarkably similar to HeatSnap+Rank, displaying a mere 2% disparity in the 7zip task, which presents the most significant difference. Notwithstanding this, our method's temporal overhead is notably lower than HeatSnap+Rank, with the gap widening as memory size increases. At a memory size of 2G, our method's temporal overhead registers at 55ms, just 57.2% of HeatSnap+Rank's overhead. As memory size escalates to 32G, the temporal overhead

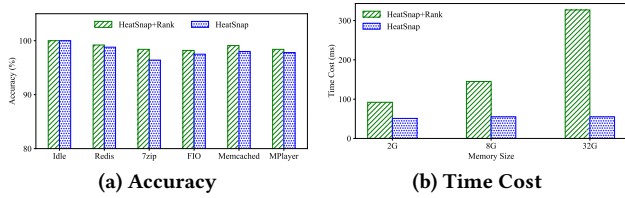

(a) Accuracy       (b) Time Cost

**Figure 4: Hot page Detection Accuracy and Time Cost**

for HEATSNAP+Rank surges to 327ms, while ours remains steadfast at 55ms – a 272ms advantage over the conventional approach. Given that the downtime averages only a few hundred milliseconds, such a discrepancy holds considerable significance.

*5.2.2 Evaluating Reaction to Hot page Transitions.* We evaluate the adaptability of our algorithm to hot page transitions through continuous snapshots. The algorithm's performance is tested during a workload shift from Memcached to FIO, focusing on the changes in hotspot detection accuracy. Snapshots are taken at 30-second intervals. For comparative analysis, we also utilize HEATSNAP+Rank to assess its effectiveness in handling hot page transitions relative to our algorithm.

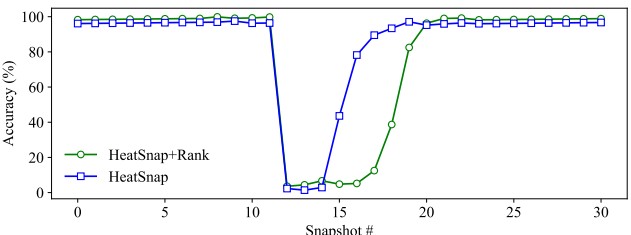

**Figure 5: Reaction to Hot page Transitions**

Figure 5 shows the temporal changes in hotspot detection accuracy for both our method and the traditional algorithm. Under a Memcached load, both approaches maintain high accuracy. However, as load changes and hotspot transitions occur, accuracy declines significantly for both. Our method quickly recovers to a reliable accuracy level within 5 samples, while the traditional technique takes 8 samples to reach a comparable accuracy level.

*5.2.3 Evaluating Dynamically Incremental Strategy.* This section evaluates our proposed dynamic incremental strategy, focusing on its feasibility and potential performance trade-offs. The analysis includes continuous snapshots taken bi-minutely across three scenarios: idle, 7zip, and a combination of CPU spin and 7zip. We define usability as $\log_{10}\left(100 \cdot \frac{N_{\text{flush}}}{N_{\text{dirty}}}\right)$, where $N_{\text{flush}}$ is the number of times dirty data is flushed to disk during the snapshot interval, and $N_{\text{dirty}}$ is the number of dirty pages in the same timeframe. A maximum usability value of 2 corresponds to copy-on-write (COW), while a value below zero indicates minimal retention of dirty data. We characterize performance loss as the ratio of additional time cost to the standard execution time cost.

Figure 6a illustrates the availability of flushing dirty data to disk over a 30-second duration under various workloads. In the idle

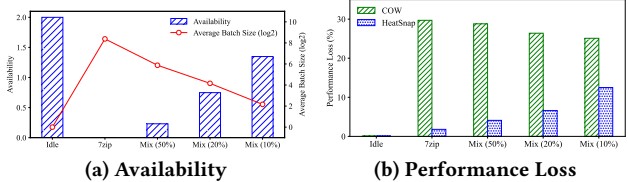

(a) Availability       (b) Performance Loss

**Figure 6: Dynamically Incremental Strategy Performance**

workload condition, availability reaches 2, indicating substantial flushing frequency. However, under the 7zip workload, availability falls below zero. This is due to the virtual machine's extreme idleness during the idle workload, prompting our dynamic incremental strategy to downgrade to the copy-on-write (COW) method, where every write operation is retained to ensure satisfactory real-time snapshot performance. In contrast, under the 7zip workload, the high activity level leads our strategy to revert to a lazily incremental approach, skipping flushing dirty data between snapshots. In a mixed scenario, the number of flushes decreases as the proportion of 7zip increases. Specifically, as the 7zip workload rises from 10% to 50%, the flushing ratio drops from 1.35 to 0.23, resulting in an average save granularity that expands from 4.5 pages to 59 pages.

Regarding performance loss, as shown in Figure 6b, the dynamic incremental strategy defaults to COW in the idle scenario. As the workload shifts to a blend of CPU spin and 7zip, the performance loss initially increases with the 7zip proportion, then decreases, reaching a minimum at full 7zip load. This fluctuation is due to the dynamic adjustment of batch size. With a low proportion of 7zip, our strategy employs a smaller batch size for dirty page data saving to enhance real-time snapshot performance. Consequently, more resources are allocated for flushing dirty data, resulting in maximal performance loss. However, as the 7zip proportion increases and the virtual machine load rises, our algorithm significantly enlarges the batch size, providing greater CPU and I/O resources to the virtual machine and effectively reducing performance loss. In contrast, the COW method experiences greater performance loss under increased virtual machine load, peaking at 29.7% during a heavily loaded 7zip task, leading to a degraded user experience.

*5.2.4 Evaluating the Feasibility of Snapshot Compression Mechanism.* As previously discussed, improving granularity can reduce redundant data, leading to smaller snapshot sizes. Therefore, we conduct experiments to determine the optimal granularity dimension. We capture a single snapshot of virtual machines under six distinct workloads: idle, 7zip, Redis, FIO, Memcached, and MPlayer, and compute the compression ratio of dirty data across various granularities. Additionally, we analyze the overlap between different task loads by determining the percentage of memory pages in the snapshots of virtual machines running each workload that can be traced in the memory pages of other workloads. We calculate the redundancy rate of each snapshot, defined as Redundancy Rate = $\frac{\sum_{\text{redundant}}}{\sum_{\text{all}}}$, where $\sum_{\text{redundant}}$ is the count of memory pages matching data in other snapshots, and $\sum_{\text{all}}$ is the total count of memory pages.

Figure 8a illustrates that the compression ratio of dirty data increases as granularity decreases. A significant improvement is observed when granularity is reduced from 4096B to 1024B, and

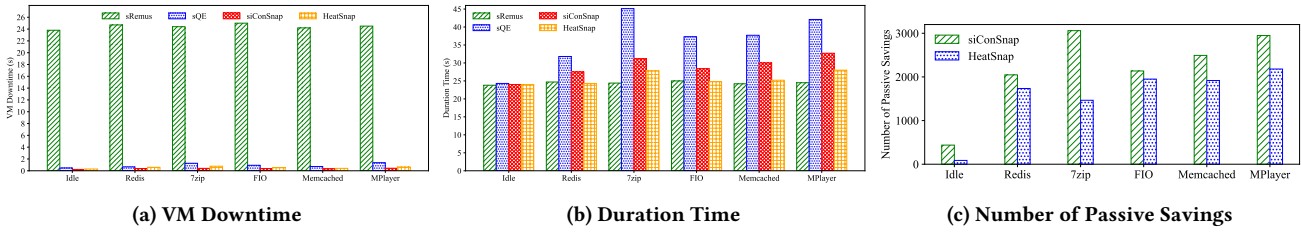

| (a) VM Downtime | (b) Duration Time | (c) Number of Passive Savings |

**Figure 7: Snapshot Metrics of Single Snapshot**

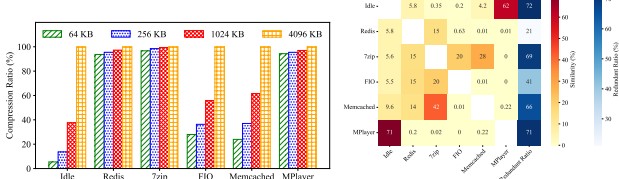

(a) Compression Ratio with Differ- (b) Redundancy Rate of Multi
ent Granularity                     Snapshots

**Figure 8: Evaluating the Feasibility of Snapshot Compression Mechanism**

further to 256B. However, the effect becomes less pronounced when granularity decreases from 256B to 64B. Thus, a granularity of 256B emerges as the most practical choice, as smaller granularities yield marginal gains in compression ratios at the cost of increased overhead.

Figure 8b presents results from multi-virtual machine experiments, revealing a high redundancy rate across most snapshot data, with over 70% of the pages being redundant. Redis shows the lowest redundancy rate at 21%, indicating that replacing redundant pages with references could significantly reduce duplicate data, enhancing compression. The analysis also highlights a high similarity rate of around 60% between Idle and MPlayer, suggesting that although MPlayer frequently modifies memory while reading video files, the number of modified pages does not increase substantially, leading to a significant overlap with the Idle state. Additionally, the similarity between the 7zip and Memcached tasks ranges from 30% to 40%, while the remaining workloads exhibit no significant similarity.

## 5.3 Macrobenchmark Performance

### 5.3.1 Single Snapshot.
In this section, we evaluate the performance of four continuous snapshot systems: sRemus, sQE, siConSnap, and HeatSnap using three metrics: *VM downtime*, *duration time*, and the *number of passive savings during post-copy*, which directly reflects performance loss.

*VM downtime.* As shown in Figure 7a, HeatSnap maintains a stable downtime of only a few milliseconds across tasks, shorter than sRemus and between sQE and siConSnap. This is due to HeatSnap's hot page detection, which requires slightly more time than siConSnap but less than sQE, which processes a larger number of pages. The downtime is stable due to the efficiency of the hotspot detection algorithm, which limits the number of pages saved during downtime.

*Duration time.* Figure 7b shows that sRemus achieves the shortest duration time as it saves all pages during downtime. HeatSnap follows, reducing duration time by 13.3% compared to siConSnap. This is attributed to efficient hot page saving during downtime and larger batch sizes for cold page saving. sQE has the longest duration due to repeated page savings. HeatSnap benefits from avoiding CPU competition between the VM and the snapshot algorithm, making its duration time comparable to sRemus even when running multiple VMs on the same machine.

*Passive savings during post-copy.* Figure 7c compares passive savings, triggered when unsaved pages are modified during snapshots. HeatSnap consistently shows 62.3% fewer passive savings than siConSnap, with only 19.9% under Idle. This reduction is due to HeatSnap's efficient selection of pages to save during downtime, reducing the need for writebacks and minimizing page faults, leading to better user experience.

### 5.3.2 Continuous Snapshots.
In this section, we evaluate the performance of four continuous snapshot systems: sRemus, sQE, siConSnap, and HeatSnap using 60 continuous snapshots taken at 30-second intervals. The performance is quantified using three metrics: *VM downtime*, *duration time*, and the *number of passive savings during post-copy*. The results are averaged across all snapshots. For consistency, the dynamic incremental strategy was deactivated, and only the lazily incremental method, similar to siConSnap, was employed.

*VM downtime.* Figure 9a shows that the *VM downtime* behavior for continuous snapshots closely matches that of single snapshots. sRemus exhibits the longest downtime, followed by sQE, with siConSnap showing the shortest downtime. HeatSnap falls between sQE and siConSnap. However, compared to single snapshots, the downtime of HeatSnap relative to siConSnap increases noticeably.

*Duration time.* As seen in Figure 9b, the results for *duration time* also align with the single snapshot scenario. sRemus shows the shortest duration, followed by HeatSnap, siConSnap, and sQE. In continuous snapshots, HeatSnap's duration time is, on average, 17.9% shorter than siConSnap's, marking an additional 4.6% reduction compared to the single snapshot case.

*Number of passive savings during post-copy.* Figure 9c reveals that HeatSnap records significantly fewer passive savings than siConSnap across different workloads, averaging 51.6% of siConSnap's count, a 10.7% improvement over the single snapshot scenario.

These improvements stem from focusing on hot pages that experience frequent modifications during downtime. In continuous snapshots, hot pages represent a larger proportion of the workload, leading to more pre-saved pages. While this slightly increases VM

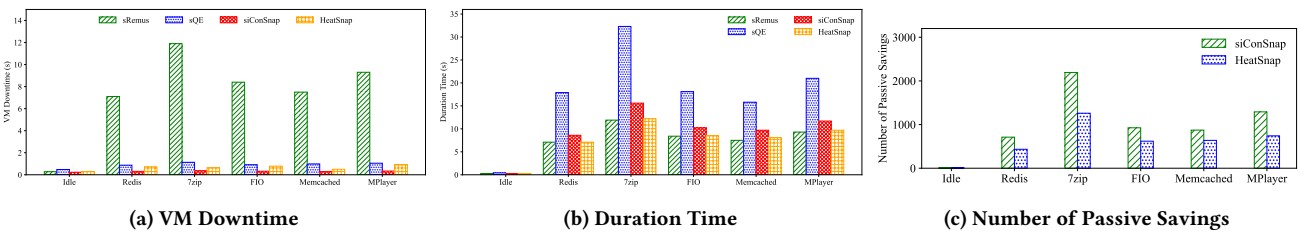

**Figure 9: Snapshot Metrics of Continuous Snapshots with 30 s interval**

downtime, it results in a notable reduction in both the *duration time* and *passive savings*, thus minimizing performance overhead.

*5.3.3 Storage Costs.* This section examines two techniques of the snapshot compression mechanism: fine-grained extraction and redundancy merging. The former is conducted employing the optimal granularity of 256B, as established in §5.2.5. The aim is to evaluate the extent to which these methods can optimize the compression for storage occupancy of snapshot files. The experiments are performed on virtual machine snapshots under various workloads comprising Idle, 7zip, Redis, FIO, Memcached, and MPlayer.

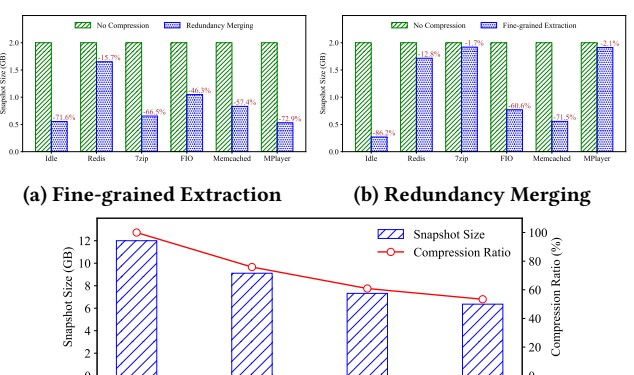

**(a) Fine-grained Extraction**    **(b) Redundancy Merging**

**(c) Redundancy Merging and Fine-grained Extraction**

**Figure 10: Snapshot Compression Mechanism Performance**

Figure 10a shows that the fine-grained extraction method performs well under idle, FIO, and Memcached workloads, yielding compression rates of 13.8%, 28.5%, and 39.4%, respectively. In contrast, other workloads exhibit lower compression due to significant spatial locality in memory page modifications, where most of the data in a dirty page is modified. Overall, fine-grained extraction effectively reduces redundant data in continuous snapshots, achieving an average compression rate of 60.9% over 12GB of memory across 6 snapshots.

Figure 10b demonstrates that redundancy merging provides high compression rates across most workloads, with the exception of Redis. Other workloads maintain approximately a 50% compression rate, with idle, 7zip, and MPlayer workloads showing around 30%. In summary, redundancy merging effectively compresses snapshot data, reaching an aggregate compression rate of 75.9% over 6 snapshot files, with the compression rate increasing as more snapshots are added.

Finally, as shown in Figure 10c, combining both methods further compresses redundant data, yielding a 53.4% compression rate for 6 snapshots. This demonstrates that nearly half of the redundant data is eliminated, underscoring the effectiveness of these techniques in minimizing redundant snapshot data.

## 6 Related Works

**Hot page Identification.** Shuang Wu et al. [36] proposed a method for hybrid-copy migration of hot pages in cloud computing using an LRU-based algorithm for hotspot detection. MLLM [28] introduced a technique that utilizes pre-copy or post-copy migration based on the hot and cold characteristics of data pages, employing a CLOCK algorithm to monitor page temperature changes. Decongest [35] created a detection algorithm that uses a write access counter to identify hot pages by setting a threshold.

**Snapshot Techniques.** FVMM [32] enhanced traditional stop-copy and pre-copy methods with template technology to optimize migration time. For continuous snapshots, existing methods often use incremental snapshots to minimize redundant data. iConSnap [13] proposed a lazy incremental saving approach that postpones incremental saving until each snapshot is created, sacrificing real-time performance for better virtual machine efficiency.

**Storage Compression.** Snapshot storage can consume substantial space. Syed Zahed K. et al. [38] introduced an rsync algorithm that segments data into blocks and generates signatures to eliminate redundancy. iConSnap [13] addresses the growth of incremental snapshot data by reducing the granularity of early snapshots with lower access probabilities. The SnapStore [23] algorithm segments memory regions to streamline deduplication based on program memory mapping structure.

## 7 Conclusion

This study introduces HEATSNAP, an effective continuous snapshot system that exploits the characteristics of hot and cold pages in virtual machine memory access patterns to expedite snapshot saving process. HEATSNAP employs a dynamic hot spot detection algorithm to pinpoint hot spot pages. For continuous snapshots, HEATSNAP utilizes a dynamically incremental strategy to delicately balance the operational efficiency of virtual machines with the availability of continuous snapshots. Furthermore, HEATSNAP presents two snapshot compression techniques (fine-grained extraction and redundancy merging) that significantly reduce the storage overhead of snapshots.

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
