# OpenReview forum: "HeatSnap: A Hot Page-Aware Continuous Snapshots System for Virtual Machines in Web Infrastructure"
_ACM.org/TheWebConf/2025/Conference — WWW 2025 Poster_

### Official Review · Reviewer_XXFL · 2024-11-25

**Novelty:** 4
**Technical Quality:** 4

**Review:**

The method seems to perform quite better than existing methods. At least it's a different and better way to do the same thing.
Looks like they have done extensive testing, from microbenchmarks to actual machine tests. The test methods seem legit.
The mechanism introduced in the article appears to be relatively straightforward and not complex overall (based on the introduction in Chapter 3), so my question is why existing work appears to be significantly inferior in experimental data?

**Questions:**

1. Chapter 3.2.2 " Initial Threshold for Hot page Identification. The initial threshold is set at 9, meaning that pages with a weight of 9 or higher are classified as hot pages". Why is it hot with a weight of 9 or higher? By experience or statistics?
2. There are similar issues in section 3.2.3. Why is this mechanism based on simple rules effective? Can others have similar works for comparison?
3. Chaper 4. The current released QEMU version is 9.x.x. But the paper chosed 2.5.0 as the research platform. Could you explain the reason?
4. Chaper 5. I cann't find any specific information about CPU models for evaluations, such as whether it is an Intel/AMD/ARM CPU, how many cores it has, and how many threads it supports. Is the experimental conclusion of the paper completely unrelated to CPU models?

**Reviewer Confidence:**

2: The reviewer is willing to defend the evaluation, but it is likely that the reviewer did not understand parts of the paper

**Scope:**

3: The work is somewhat relevant to the Web and to the track, and is of narrow interest to a sub-community

---

### Official Review · Reviewer_x9RY · 2024-11-25

**Novelty:** 5
**Technical Quality:** 5

**Review:**

The paper addresses VM snapshot issues, including downtime, performance loss, and snapshot size. Its relevance to Web infrastructure lies in its focus on large-scale web environments. HeatSnap is introduced to reduce VM downtime and snapshot size during snapshot creation by leveraging the characteristics of hot pages. The main contribution of this paper is the classification of pages into hot and cold, applying different snapshot creation methods to each, which is innovative.

The introduction clearly outlines existing VM snapshot methods, highlighting their advantages and limitations. The background section thoroughly explains the characteristics and challenges of hot pages, providing the theoretical basis for HeatSnap’s optimization of the VM snapshot process. The evaluation section compares HeatSnap with existing continuous snapshot systems, demonstrating its advantages. However, a sensitivity analysis of the threshold for classifying hot and cold pages would improve the evaluation and deepen insights into the method's robustness.

In conclusion, the writing is well-done in the background and introduction sections, but some design details are not sufficiently elaborated, as noted in the "Questions" section. While the paper aims to address Web infrastructure challenges, it lacks an end-to-end Web workload test. Specifically, it does not evaluate a real-world Web application that depends on components like databases or memcached, deployed on VMs. Instead, the workload tested is a single application, which is not even a web application. Another critical issue lies in the figures: the text in the figures is too small to read, creating obstacles to understanding.

**Questions:**

1. In Section 3.2.1, the paper mentions weights. Does the first weight correspond to the most recently modified page?
2. In the example in Section 3.2.3, the working set weights are listed as 1, 2, 3, 8, 8. However, according to the description in Section 3.2.1, the weights should be 1, 2, 4, 8, 16. The authors did not explain this discrepancy clearly.
3. Is the title of Section 5.3, "Macrobenchmark," a typo? Should it be "Microbenchmark"?
4. In Figures 7(c) and 9(c), why does HeatSnap only compare to siConSnap?
5. Could you explain why HeatSnap is faster than HeatSnap+Rank? It seems HeatSnap+Rank uses a simpler algorithm.

**Reviewer Confidence:**

2: The reviewer is willing to defend the evaluation, but it is likely that the reviewer did not understand parts of the paper

**Scope:**

2: The connection to the Web is incidental, e.g., use of Web data or API

---

### Official Review · Reviewer_6wcH · 2024-11-30

**Novelty:** 3
**Technical Quality:** 2

**Review:**

The paper proposes the HeatSnap system, offering an innovative solution for virtual machine snapshot optimization, with significant performance and storage efficiency benefits demonstrated through experiments. The method has strong practical value, especially in cloud computing and large-scale virtualization environments. However, the paper has some weaknesses in terms of methodological detail, resource consumption analysis, and experimental setup. I recommend that the authors address these issues by providing more thorough explanations of the trade-offs, adding comparisons with existing methods, and improving the formatting and clarity of figures.
1.While the paper mentions optimizing snapshots of "hot" and "cold" pages to improve efficiency, it does not clarify how snapshot data integrity and accuracy are maintained while reducing storage overhead. Is there a trade-off between snapshot precision and efficiency? The authors should further clarify this aspect and explain how precision is maintained in different scenarios.
2.The paper focuses on the system's performance benefits, but it does not discuss the resource consumption cost, particularly the impact on host machine memory usage.
3.It seems that this paper has some formatting issues, such as inconsistent capitalization and improper symbol usage. Additionally, the figures do not clearly reflect the differences in experimental results.

**Questions:**

1.As a hypervisor-layer snapshot optimization, the paper does not consider the impact of the complex prediction and optimization algorithms on the host machine’s memory usage?
2. The Redundancy Merging strategy seems to require running multiple similar VMs on the same host machine to be effective? However, if each VM uses different snapshot cycles and strategies, will this still be effective?
Presentation problem：
1. The vertical axis scale in Figures 7a and 9a is too small, making it hard to see the data. The authors should adjust the scale to make the bar graphs clearer and ensure that the differences between similar values are distinguishable.
2.The indentation of the first line in Related Works section is inconsistent. The authors should standardize the formatting to improve the paper’s presentation.
3. In Figure 2, the "hot page detection" section uses three colors, but only red and blue are described. What does the yellow color represent? The authors should provide a clear explanation of the colors and their meanings.
4. As an overview, Figure 2 lacks clear connections between components. Adding arrows between layers to indicate the flow of the process would better illustrate the relationships between the components and improve the clarity of the diagram.
5.There are several formatting issues in the paper, such as capitalization and symbol problems. For example, in section 5.2.1, the third line has “as```.where.” The authors should ensure consistent formatting throughout the paper.

**Reviewer Confidence:**

3: The reviewer is confident but not certain that the evaluation is correct

**Scope:**

3: The work is somewhat relevant to the Web and to the track, and is of narrow interest to a sub-community

---

### Official Review · Reviewer_U6uz · 2024-12-02

**Novelty:** 5
**Technical Quality:** 5

**Review:**

## Summary:

This paper presents HeatSnap, a continuous snapshot system for virtual machines that leverages hot page characteristics to optimize snapshot performance. The system introduces novel approaches for hot page detection, dynamic incremental snapshots, and snapshot compression. The authors implemented a prototype on QEMU/KVM and demonstrated improvements in snapshot duration (17.4% reduction), VM performance loss (48.4% reduction), and storage costs (46.7% reduction) compared to existing methods.

## Strengths:
- Novel approach incorporating hot page awareness into VM snapshots
- Comprehensive system design addressing multiple aspects (detection, storage, compression)
- Clear performance improvements over existing solutions
- Practical implementation on widely-used QEMU/KVM platform
- Thorough evaluation using diverse workloads

## Weaknesses:
- Limited theoretical analysis of the hot page detection algorithm
- Lacks discussion of system limitations and failure scenarios
- Missing comparison with some recent snapshot solutions
- Insufficient scalability evaluation for large-scale deployments
- Limited discussion of memory overhead

## Detailed Comments:

### Originality:
The hot page-aware approach is novel, but the paper should better position it against existing memory management techniques that consider page access patterns.
The dynamic threshold adjustment mechanism needs more theoretical justification for its effectiveness.
### Importance of Contribution:
While the performance improvements are significant, the paper should better quantify the real-world impact on cloud infrastructure costs and management.
The broader applicability of the approach to other virtualization platforms is not adequately discussed.
### Soundness:
The hot page detection threshold of 9 seems arbitrary - the paper should provide better justification for this choice.
The paper lacks analysis of the trade-offs between snapshot frequency and system overhead.
The impact of different workload patterns on the effectiveness of the dynamic incremental strategy is not thoroughly examined.
The paper doesn't address potential race conditions in the hot page detection mechanism.
### Evaluation:
The experimental setup details are insufficient - missing information about hardware configurations and workload parameters.
The evaluation lacks stress testing under extreme conditions (e.g., memory pressure, high I/O load).
The paper should include more diverse workloads, particularly those representing modern cloud-native applications.
Missing evaluation of system behavior during concurrent snapshots of multiple VMs.

**Questions:**

- Could you provide more detailed justification for choosing this initial value, and explain how sensitive your system's performance is to different threshold values? It would be helpful to see experimental results showing system behavior across a range of initial thresholds.
- The paper demonstrates impressive performance improvements with common workloads, but how does HeatSnap perform in more challenging scenarios, such as when managing concurrent snapshots of multiple VMs with memory-intensive workloads? Could you provide data on the system's scalability limits and resource consumption patterns in such scenarios?

**Reviewer Confidence:**

2: The reviewer is willing to defend the evaluation, but it is likely that the reviewer did not understand parts of the paper

**Scope:**

3: The work is somewhat relevant to the Web and to the track, and is of narrow interest to a sub-community

---

### Official Review · Reviewer_ZPkB · 2024-12-02

**Novelty:** 6
**Technical Quality:** 7

**Review:**

The paper presents a novel and practical solution for optimizing continuous snapshots in virtualized environments through HeatSnap, a hot page-aware system. The focus on leveraging memory access patterns (hot and cold pages) to reduce performance bottlenecks and storage overheads is innovative and addresses a pressing challenge in virtual machine management for cloud infrastructure.

### **Strengths:**
1.	Novelty: The introduction of hot page-aware mechanisms, dynamic incremental strategies, and lossless snapshot compression techniques represents significant advancements over traditional snapshotting methods.
2.	Performance: The evaluation demonstrates measurable improvements in key metrics (e.g., 48.4% reduction in performance loss, 17.4% shorter snapshot duration).
3.	Clarity: The paper is well-structured, with clear explanations of methodologies, and experimental setups.
4.	Significance: By addressing scalability, performance, and storage challenges, HeatSnap is highly relevant for web-scale infrastructures and continuous data protection.

### **Weaknesses and Areas for Improvement:**
1.	Limited Discussion on Scalability: The paper does not provide detailed insights into how HeatSnap would scale in large, multi-tenant environments.
2.	Runtime Overhead: While performance gains are highlighted, the overhead of hot page detection and compression strategies is not deeply explored.
3.	Comparative Baselines: Only a few baseline systems are used for comparison; additional baselines could strengthen the results.

**Questions:**

1.	How does HeatSnap handle workloads with highly irregular memory access patterns? Are there scenarios where the system might fail to identify hot pages effectively?
2.	Could the proposed dynamic incremental strategy introduce latency spikes under unpredictable workloads? If so, how are these managed?
3.	How does the runtime overhead of HeatSnap’s compression mechanism compare with existing methods across large-scale environments?
4.	What are the implications of using HeatSnap in environments with strict real-time requirements?

**Reviewer Confidence:**

4: The reviewer is certain that the evaluation is correct and very familiar with the relevant literature

**Scope:**

4: The work is relevant to the Web and to the track, and is of broad interest to the community